# Natural Image Manipulation for Autoregressive Models Using Fisher Scores

## Abstract

Deep autoregressive models are one of the most powerful models that exist today which achieve state-of-the-art bits per dim. However, they lie at a strict disadvantage when it comes to controlled sample generation compared to latent variable models. Latent variable models such as VAEs and normalizing flows allow meaningful semantic manipulations in latent space, which autoregressive models do not have. In this paper, we propose using Fisher scores as a method to extract embeddings from an autoregressive model to use for interpolation and show that our method provides more meaningful sample manipulation compared to alternate embeddings such as network activations.

## 1 Introduction

Over the last few decades, unsupervised learning has been a rapidly growing field, with the development of more complex and better probabilistic density models. Autoregressive generative models (Salimans et al., 2017; Oord et al., 2016a; Menick & Kalchbrenner, 2018) are one of the most powerful generative models to date, as they generally achieve the best bits per dim compared to other likelihood-based models such as Normalizing Flows or Variational Auto-encoders (VAEs) (Kingma & Welling, 2013; Dinh et al., 2016; Kingma & Dhariwal, 2018). However, it remains a difficult problem to perform any kind of controlled sample generation using autoregressive models. For example, flow models and VAEs are structured as latent variable models and allow meaningful manipulations in latent space, which can then be mapped back to original data distribution to produce generated samples either through an invertible map or a learned decoder.

In the context of natural image modelling, since discrete autoregressive models do not have continuous latent spaces, there is no natural method to apply controlled generation. When a latent space is not used, prior works generally perform controlled sample generation through training models conditioned on auxiliary information, such as class labels or facial attributes (Van den Oord et al., 2016). However, this requires a new conditional model to be trained for every new set of labels or features we want to manipulate, which is a time-consuming and tedious task. Ideally, we could structure an unconditional latent space that the autoregressive model could sample from, but where would this latent space come from?

In this paper, we propose a method of image interpolation and manipulation in a latent space defined by the Fisher score of both discrete and continuous autoregressive models. We use PixelCNNs to model the natural image distribution and use them to compute Fisher scores. In order to map

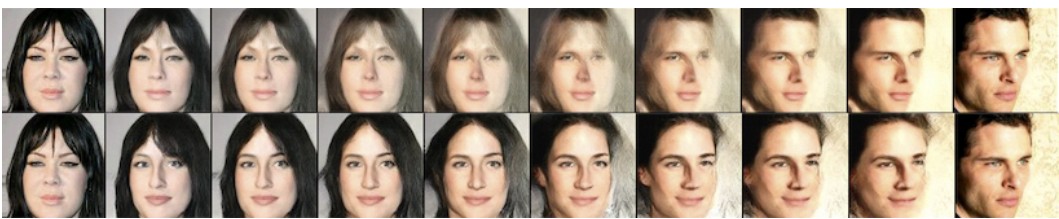

Figure 1: CelebA interpolation comparing PixelCNN activations (top row) versus Fisher scores (bottom row) as embedding spaces for autoregressive models

back from Fisher score space to natural images, we train a decoder by minimizing reconstruction error. We show that interpolations in Fisher score space provide higher-level semantic meaning compared to baselines such as interpolations in PixelCNN activation space, which produce blurry and incoherent intermediate interpolations similar in nature to interpolations using pixel values. In order to evaluate interpolations quantitatively, for different mixing coefficients $\alpha$, we calculate FID (Heusel et al., 2017) of the images decoded from a large sample of convex combinations of latent vectors.

In summary, we present two key contributions in our paper:

- A novel method for natural image interpolation and semantic manipulation using autoregressive models through Fisher scores

- A new quantitative approach to evaluate interpolation quality of images

## 2 RELATED WORK

There exists a substantial amount of work on natural image manipulation using deep generative models. VAEs (Higgins et al., 2017), BigBiGANS (Donahue et al., 2016; Donahue & Simonyan, 2019; Brock et al., 2018) and normalizing flows Kingma & Dhariwal (2018) provide learned latent spaces in which realistic image manipulation is a natural task. Interpolating through a latent space allows more natural transitions between images compared to pixel value interpolations, which naively overlay images on top of each other. Other prior methods learn hierarchical latent spaces on GANs and VAEs to encourage semantic manipulations to disentangle different levels of global characteristics of facial features, such as skin color, gender, hair color, and facial features (Karras et al., 2019).

Aside from using a latent space for controlled image generation, prior methods have also trained generative methods conditioned on relevant auxiliary feature labels, such as class labels in ImageNet. Other prior works have also used facial feature embeddings and binary facial attributes from CelebA to manipulate facial characteristics of generated images (Van den Oord et al., 2016).

Similarly, there has been a large amount of work on using Fisher information in machine learning. Many prior methods use the Fisher Kernel in algorithms such as kernel discriminant analysis and kernel support vector machines (Mika et al., 1999; Jaakkola & Haussler, 1999). More recent works introduce the Fisher vector, which uses Fisher scores of mixtures of Gaussians as feature vectors for downstream tasks (Sánchez et al., 2013; Simonyan et al., 2013; Perronnin et al., 2010b; Gosselin et al., 2014; Perronnin et al., 2010a). However, to our knowledge, there has been no work in using Fisher scores for deep generative modelling.

## 3 BACKGROUND

### 3.1 PIXELCNN

PixelCNNs (Oord et al., 2016b) are powerful autoregressive models used to model complex image distributions. They are likelihood-based generative models that directly model the data distribution $p(x)$. This allows us to exactly compute the Fisher score instead of using an approximation, which would be necessary for GANs (Goodfellow et al., 2014) or VAEs, as they either implicitly optimize likelihood, or optimize a variational lower bound.

PixelCNNs use a series of masked convolutions to define an autoregressive model over image data. Masked convolutions allow PixelCNNs to retain the autoregressive ordering when propagating values through layers. Over the past few years, many improved variants of PixelCNNs have been developed to address certain problems with the original PixelCNN design. Specifically for our work, we use Gated PixelCNNs (Van den Oord et al., 2016), which introduced horizontal and vertical convolutional blocks to remove the blind-spot in the original masked convolutions of PixelCNNs, and Pyramid PixelCNNs (Kolesnikov & Lampert, 2017), which use PixelCNN++ (Salimans et al., 2017) architectures in a hierarchical fashion, with each layer of the hierarchy modelling images at different down-sampled resolutions conditioned on the previous layer.

## 3.2 FISHER SCORE

The Fisher score $\dot{\ell}(x;\theta) = \nabla_\theta \log p_\theta(x)$ is defined as the gradient of the log-probability of a sample with respect to the model parameters $\theta$. Intuitively, Fisher scores describe the contributions of each parameter during the generation process. Similar samples in the data distribution should elicit similar Fisher scores. Similarity between samples can also be evaluated by taking a gradient step for one sample, and checking if the log-likelihood of another sample also increases. This intuition is re-affirmed in the underlying mathematical interpretation of Fisher information - the Fisher score maps data points onto a Riemannian manifold with a local metric given by the Fisher information matrix. The underlying kernel of this space is the Fisher kernel, defined as:

$$\dot{\ell}(x_i;\theta)^T F^{-1} \dot{\ell}(x_j;\theta) = (F^{-\frac{1}{2}} \dot{\ell}(x_i;\theta))^T (F^{-\frac{1}{2}} \dot{\ell}(x_j;\theta))$$

where $F^{-1}$ is the inverse Fisher information matrix, and $F^{-\frac{1}{2}}$ is the Cholesky Decomposition. Applying $F^{-\frac{1}{2}}$ is a normalization process, so the Fisher kernel can be approximated by taking the dot product of standardized Fisher scores. This is useful since computing $F^{-1}$ is normally difficult. As such, we can see the collection of Fisher scores as a high dimensional embedding space in which more meaningful information about the data distribution can be extracted compared to raw pixel values. More complex deep generative models may learn parameters that encode information at high-levels of abstraction which may be reflected as high-level features in Fisher scores.

## 3.3 SPARSE RANDOM PROJECTIONS

It would be cumbersome to work in the very high-dimensional parameter spaces of deep generative models, so we use dimensionality reduction methods to make our methods more scalable. Sparse random projections allow for memory-efficient and scalable projections of high dimensional vectors. The *Johnson-Lindenstrauss Lemma* (Dasgupta & Gupta, 1999) states that under a suitable orthogonal projection, a set of $n$ points in a $d$-dimensional space can be accurately embedded to some $k$-dimensional vector space, where $k$ depends only on $\log n$. Therefore, for suitably large $k$, we can preserve the norms and relative distances between projected points, even for very high dimensional data. Since sparse random matrices are nearly orthogonal in high-dimensional settings, we can safely substantially reduce the dimensionality of our embedding spaces using this method.

We generate sparse random matrices according to Li et al. (2006). Given an $n \times k$ matrix to project, we define the minimum density of our sparse random matrix as $d = \frac{1}{\sqrt{k}}$, and let $s = \frac{1}{d}$. Suppose that we are projecting the data into $p$ dimensions, then the $n \times p$ projection matrix $P$ is generated according to the following distribution:

$$P_{ij} = \begin{cases} -\frac{s}{\sqrt{p}} & \text{with probability } \frac{1}{2s} \\ 0 & \text{with probability } 1 - \frac{1}{s} \\ \frac{s}{\sqrt{p}} & \text{with probability } \frac{1}{2s} \end{cases} \tag{1}$$

where $P_{ij}$ is the element of the $i$th row and $j$th column of the projection matrix.

---

**Algorithm 1:** The procedural generation of the new interpolated embedding dataset for quantitative evaluation. Following FID conventions, we use a sample size of 50000 images.

---

**Result:** $\alpha$-interpolated dataset $\mathcal{D}_\alpha$
**Input:** The dataset $\mathcal{D}$, projection matrix $P$, pre-trained autoregressive model $p_\theta(x)$, and the learned decoder $Dec$
$\mathcal{D}_\alpha \leftarrow \{\}$
**for** $i \leftarrow 0$ **to** $50000$ **do**
    Sample $x_1, x_2 \sim \mathcal{D}$, a random pair of samples
    $z_1 \leftarrow P\nabla_\theta \log p_\theta(x_1)$
    $z_2 \leftarrow P\nabla_\theta \log p_\theta(x_2)$
    $\hat{z} \leftarrow (1-\alpha)z_1 + \alpha z_2$
    $\hat{x} \leftarrow Dec(\hat{z})$
    $\mathcal{D}_\alpha \leftarrow \mathcal{D}_\alpha \cup \{\hat{x}\}$
**end**

---

## 4 METHOD

We now describe our approach for natural image manipulation and interpolation using autoregressive models. Note that this method is not restricted to autoregressive models and can be applied to any likelihood-based models.

### 4.1 EMBEDDING SPACE CONSTRUCTION

Given a trained autoregressive model, $p_\theta(x)$, we want to construct an embedding space that is more meaningful than raw pixel values. In this paper, our autoregressive models are exclusively variants of PixelCNNs since we are working in an image domain. Drawing inspiration off popular self-supervised methods (Zhang et al., 2016; Gidaris et al., 2018; Doersch et al., 2015; Pathak et al., 2016; Noroozi & Favaro, 2016), it may seem natural to take the output activations of one of the last few convolutional layers. However in our experiments, we show that using output activations provides *less* meaningful embedding manipulation than using the Fisher score. This is especially the case for PixelCNNs, as PixelCNNs are known for generally encoding more local statistics than global features of images. However, Fisher scores lie in parameter space, which may encode more global semantics.

In order to construct the embedding space using Fisher scores, we initialize a sparse random matrix $P$, randomly generated following the distribution in equation 1. In practice, we found sparse random matrices the only feasible method for reasonable dimensionality reduction when scaling to more difficult datasets, where the sizes of corresponding PixelCNNs grew to tens of millions of parameters. Finally, the embedding space is constructed as follows: each element $z_i$ in the new embedding space is computed as $z_i = P\nabla_\theta \log p_\theta(x_i)$ for each sample $x_i$ in the dataset.

### 4.2 LEARNING A DECODER

Regardless of whether we use Fisher scores or network activations as embedding spaces, doing any sort of image manipulation in a generated embedding space requires a mapping back from $z_i$ to $x_i$. To solve this problem, we learn a mapping back from $z_i$ to $x_i$ by training a network to model the density $p(x_i|z_i)$. We try both supervised and unsupervised learning approaches, either directly learning a decoding model to minimize reconstruction error, or implicitly learn reconstruction by training a conditional generative model, such as another autoregressive model or a flow.

### 4.3 INTERPOLATION EVAUATION METRIC

We introduce an evaluation procedure to quantitatively evaluate image interpolation quality. The quality of image interpolations can roughly be measured by how realistic any intermediate interpolated image is. Existing popular methods to measure image quality can largely be attributed GAN metrics, such as Fréchet Inception Distance (FID) (Heusel et al., 2017) or Inception Score (IS) (Salimans et al., 2016). We choose to use FID as our evaluation metric since it is more generalizable to other datasets than IS. Interpolation quality is evaluated by computing FID for different mixing coefficients $\alpha$. For $\alpha \in \{0, 0.125, \ldots, 0.5\}$, we generate a new dataset $\mathcal{D}_\alpha$ according to Algorithm 1, and compute the FID between the true dataset and $\mathcal{D}_\alpha$. Under this evaluation metric, good interpolations result in lower FID scores, and bad interpolations produce peaked FID scores at $\alpha = 0.5$.

## 5 EXPERIMENTS

We evaluate our method on MNIST and CelebA and design experiments to answer the following questions:

- How does image interpolation quality using Fisher scores compare to baseline embedding spaces such as PixelCNN activations?
- Do Fisher scores contain high-level semantic information about the original image data?

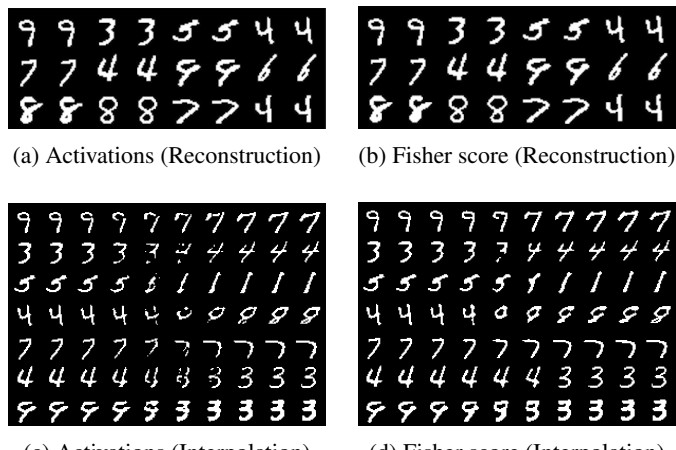

(a) Activations (Reconstruction)     (b) Fisher score (Reconstruction)

(c) Activations (Interpolation)     (d) Fisher score (Interpolation)

Figure 2: Comparison between between using network activations and Fisher scores as underlying embedding spaces for reconstruction and interpolation. (a) and (b) show reconstructions, where for each image pair, the left image is the true image, and the right image is the reconstruction. For (c) and (d), each row shows an interpolation from one image in the MNIST dataset to another.

|      | 0.001 | 0.01 | 0.1 | 1.0 |
|------|-------|------|-----|-----|
| 64   | 0.641 | 0.638 | 0.678 | 0.633 |
| 256  | 0.653 | 0.456 | 0.170 | 0.155 |
| 1024 | 0.142 | 0.154 | 0.129 | 0.122 |

(a)

| $\alpha =$ | 0.0 | 0.125 | 0.25 | 0.375 | 0.5 |
|------|-----|-------|------|-------|-----|
| Activations | 0.68 | 0.60 | 2.69 | 12.84 | 25.42 |
| **Fisher Score (ours)** | **1.21** | **1.14** | **1.03** | **2.21** | **5.43** |

(b)

Figure 3: (a) shows reconstruction error on our decoder for different different combinations of density and projection dimensions for sparse matrices. The top row is density, and the left column is projection dimension. (b) compares the FID between activations and Fisher scores as embedding spaces for increasing levels of interpolation. We can see that FID values for our method are much more stable compared to the baseline

## 5.1 MNIST

**Setup** We trained a standard PixelCNN with masked convolutions on binarized MNIST images as our autoregressive model. Images were binarized by sampling from Bernoulli random variables biased by grayscale pixel values. The decoder model consisted of a dense layer following by 3 transposed convolutional layers that upscale the image to $28 \times 28$.

**Projection Method** We primarily experimented with varying levels of density and projection dimensions for dense and sparse random projections. Figure 3 (a) shows reconstruction error on trained decoders for different hyperparameter combinations. For MNIST, we chose to use a dense matrix with a projection dimension of 1024.

**Results** Figure 2 shows a comparison between using the PixelCNN activations of the second-to-last convolutional layer and projected Fisher scores as embedding spaces. In general, reconstruction using the activations is slightly more accurate than using Fisher scores. However, for interpolations, using activations produces intermediate images that are similar to interpolations using raw pixel values. On the other hand, interpolations with our method using Fisher scores produces a more natural transition between samples. These qualitative observations are also reflected in our quantitative measurements shown in Figure 2. The interpolations using activations reaches a much higher FID (25.42) than using interpolation using Fisher scores (5.43). Note that we only show $\alpha$ from 0 to 0.5, as FID scores for $\alpha = 0.75$ are the same as $\alpha = 0.25$ since mixing coefficients are symmetric.

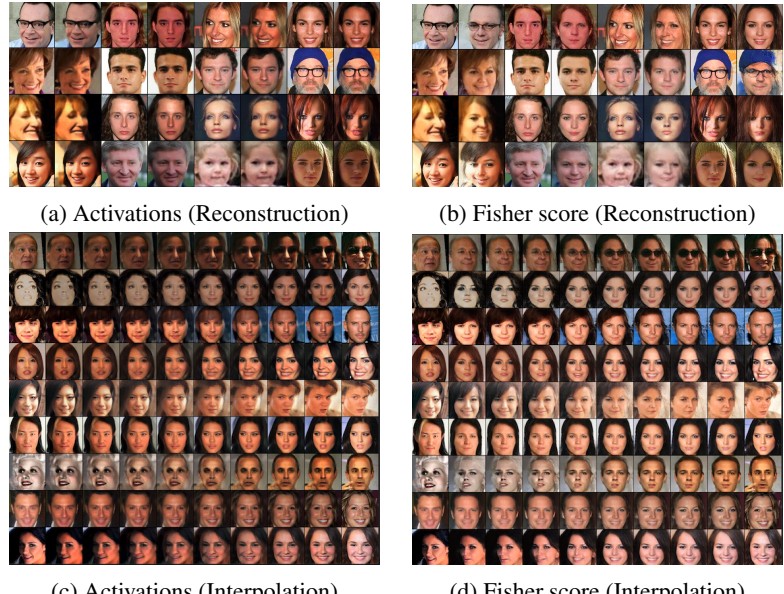

(a) Activations (Reconstruction)        (b) Fisher score (Reconstruction)

(c) Activations (Interpolation)        (d) Fisher score (Interpolation)

Figure 4: Comparison between using network activations and Fisher scores as underlying embedding spaces for reconstruction and interpolation. (a) and (b) show reconstructions, where for each image pair, the left image is the true image, and the right image is the reconstruction. For (c) and (d), each row shows an interpolation from one image in the CelebA dataset to another. The left-most and right-most of each row is the true image pair from the dataset.

## 5.2 CELEBA

**Dataset** We evaluate our method 128x128 CelebA images. We follow the standard procedure of cropping raw 218 x 178 CelebA images into 128 x 128 CelebA images. All autoregressive models and decoders are trained to model 5-bit images, since this bit reduction results in faster learning with a negligible cost in visible image quality.

**Autoregressive Architecture** In order to investigate the effect of model complexity on Fisher score representations, we experimented with three different variations of PixelCNN architectures: a standard PixelCNN with 5 layers (same as MNIST), a Gated PixelCNN, and a Pyramid PixelCNN, each with about 1 million, 5 million, and 20 million parameters respectively. We trained all models with batch size 128 and learning rate $1e-3$ using an Adam optimizer for 50 epochs. More architecture details can be found in the appendix.

**Decoder Architecture** We experimented with different kinds of decoder architectures. We tested a convolutional decoder architecture with added residual blocks, and also various conditional generative models, such as conditional RealNVPs, Pyramid PixelCNNs, and WGANs (Gulrajani et al., 2017) which helped produce less noisy reconstructions. Out of all decoder architectures, we observed that the conditional RealNVP produced the best qualitative results. See the appendix for more samples of each decoder architecture.

**Projection Method** We use sparse random matrices to project the Fisher Scores with the default density $\frac{1}{\sqrt{n_{features}}}$, and projection dimension of 16384. We then apply PCA to reduce the dimensionality of the Fisher scores to 4096. We found that using PCA has minimal degradation on reconstruction quality, and significantly reduced the number of parameters in our decoder models. Using this process showed better quality reconstruction compared to directly applying a random projection to 4096 dimensions, which suggests that the random projection is the primary bottleneck in this method.

**Reconstruction and Interpolation** Figure 4 shows a comparison between reconstruction and interpolation for PixelCNN activations versus Fisher scores. We note that since we are using

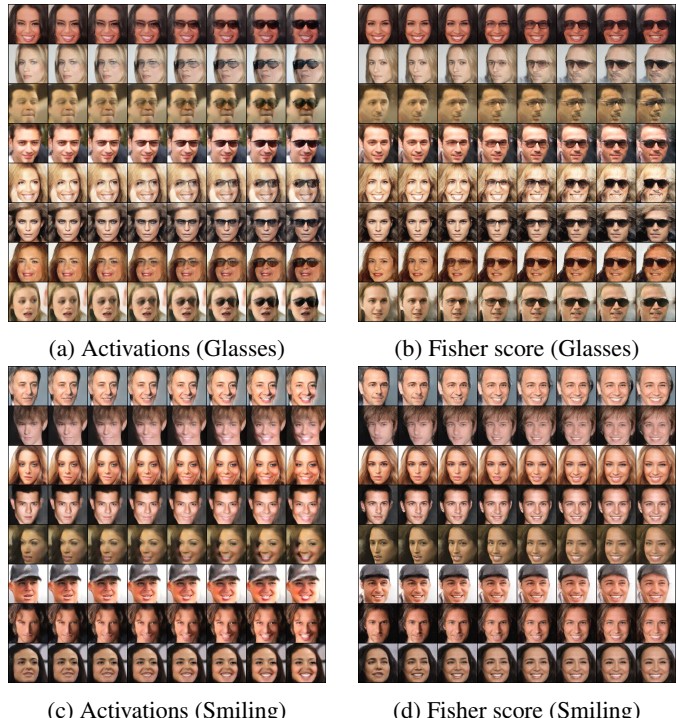

| (a) Activations (Glasses) | (b) Fisher score (Glasses) |
|---|---|
| (c) Activations (Smiling) | (d) Fisher score (Smiling) |

Figure 5: We compare semantic manipulation on glasses and smiling attributes for PixelCNN activation versus Fisher score (ours) embedding spaces

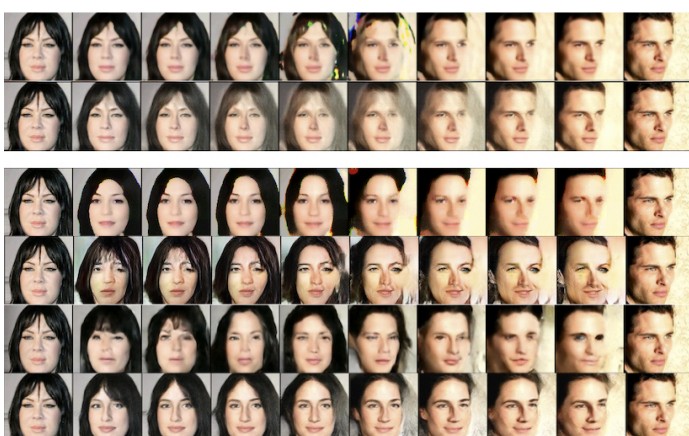

Figure 6: The top two rows show interpolation using Conv and RealNVP decoders respectively with PixelCNN activations as an embedding space. The bottom four rows show interpolations using Conv, WGAN, Pyramid PixelCNN, and RealNVP decoders with Fisher scores (our method) as an embedding space. We can see that interpolations using our method are consistent across different decoders, which supports our claim that high-level semantic information is indeed stored in the Fisher scores, and not due to the decoder architectures themselves

a decoder to reconstruct images, interpolations do not begin and end with the true images of the dataset, and instead, are approximations constructed from the decoder. We see similar results an in our MNIST experiments. Reconstruction quality for activations is better than with Fisher scores. However, interpolations for Fisher scores is qualitatively more natural than those of activations. Figure 6 shows that similar interpolation characteristics were observed across different decoder architectures. Therefore, it is unlikely that the good interpolations arise solely from the stronger decoder networks, and instead meaningful semantic information is stored in the Fisher scores

| Pyramid PixelCNN as Autoregressive Model | | | | | | |
|---|---|---|---|---|---|---|
| Decoder | Embedding | $\alpha = 0.0$ | $\alpha = 0.125$ | $\alpha = 0.25$ | $\alpha = 0.375$ | $\alpha = 0.5$ |
| Conv | Activation | 63.23 | 63.38 | 67.19 | 74.98 | 80.10 |
| | Fisher Score (ours) | 136.05 | 130.70 | 127.98 | 127.62 | 128.36 |
| RealNVP | Activation | 24.52 | 28.32 | 36.81 | 46.98 | 51.75 |
| | **Fisher Score (ours)** | **31.16** | **33.85** | **37.30** | **40.75** | **42.38** |

| Gated PixelCNN as Autoregressive Model | | | | | | |
|---|---|---|---|---|---|---|
| Decoder | Embedding | $\alpha = 0.0$ | $\alpha = 0.125$ | $\alpha = 0.25$ | $\alpha = 0.375$ | $\alpha = 0.5$ |
| Conv | Activation | 56.31 | 56.92 | 61.28 | 77.38 | 101.59 |
| | Fisher Score (ours) | 129.45 | 128.99 | 129.90 | 132.50 | 134.38 |
| RealNVP | Activation | 24.42 | 30.20 | 41.76 | 53.87 | 59.00 |
| | **Fisher Score (ours)** | **33.46** | **36.60** | **41.74** | **48.31** | **52.17** |

| PixelCNN (5 layers) as Autoregressive Model | | | | | | |
|---|---|---|---|---|---|---|
| Decoder | Embedding | $\alpha = 0.0$ | $\alpha = 0.125$ | $\alpha = 0.25$ | $\alpha = 0.375$ | $\alpha = 0.5$ |
| Conv | Activation | 54.40 | 55.75 | 61.63 | 89.38 | 127.85 |
| | Fisher Score (ours) | 132.47 | 131.85 | 132.33 | 133.76 | 134.43 |
| RealNVP | Activation | 22.29 | 26.42 | 34.41 | 42.92 | 46.99 |
| | **Fisher Score (ours)** | **30.27** | **32.14** | **35.53** | **39.65** | **41.83** |

Figure 7: Interpolation evaluation results for CelebA. We show quantitative results for varying combinations of base autoregressive models paired with different decoders. We compare using network activations versus Fisher scores (our method) as embedding spaces.

themselves. These observations are supported by the quantitative results shown in Figure 7. For the RealNVP decoder, we can see that although using network activations began with a lower FID (better reconstruction), increasing levels of interpolation result in FID rising faster to a higher peak than the same interpolation level for Fisher scores.

Looking at the effect of model complexity, we see that there is not too much of a difference in reconstruction and interpolation quality, but generally, less complex models allow slightly more accurate image reconstructions. Good interpolations for even the simpler models suggest that they are still learning a substantial amount of information about the data including high level semantic information.

**Semantic Manipulation** Since CelebA has binary labels, we can also look at semantic manipulation in addition to interpolations. Given a binary attribute, such as the presence of black hair, a smile, or eyeglasses, we can extrapolate in embedding space in the same manner described in Glow (Kingma & Dhariwal, 2018). For some attribute, let $z_{pos}$ be the average of all embedding vectors with the attribute, and $z_{neg}$ be the average of all embedding vectors without the attribute. We can then apply or remove the attribute by manipulating a given embedding vector in the direction of $\delta = z_{pos} - z_{neg}$ or its negation. For our experiments, we found that scaling $\delta$ by a factor of 3 was enough to see visible change in our images. See Figure 5 for example of applying different attributes using both the activation and Fisher score embedding spaces. Overall, our method of using Fisher scores tends to encourage more natural semantic manipulations in adding a smile, or adding eyeglasses compared to activation embeddings, which tend to "paste" generic smiles and glasses on top of each face.

## 6 CONCLUSION

We proposed a new method to allow image interpolation and manipulation using autoregressive models. Our approach used Fisher scores an as underlying embedding space, and showed more natural interpolation compared to our baselines using activations of PixelCNNs or raw pixel interpolations. In addition, we introduced a new evaluation metric for interpolations by using FID to measure images at different levels of interpolation. Lastly, we note that this method is not restricted to images, and generalizes to any autoregressive model on any kind of dataset.

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

# 7 APPENDIX

## 7.1 PIXELCNN ARCHITECTURES

### 7.1.1 MNIST

The PixelCNN architecture for MNIST was a standard PixeCNN Oord et al. (2016b), with 5 masked convolutional layers each kernel size 7, padding 3, and 64 filters. We used ReLU for our activation function.

### 7.1.2 CELEBA

**1-layer PixelCNN**  Architecture is the same as the PixelCNN from MNIST, but with only 1 masked convolutional layer.

**5-layer PixelCNN**  Architecture is the same as the PixelCNN from MNIST

**Gated PixelCNN**  The Gated PixelCNN architecture is the same as described in Van den Oord et al. (2016). We use a filter size of 120 dimensions, with 5 masked convolutional layers of kernel size 7 and padding 2. Each masked convolutional layer uses vertical and horizontal stack of convolutions, with residual connections and gating mechanisms.

**Pyramid PixelCNN**  The Pyramid PixelCNN has 3 layers. The bottom layer is a Pixel-CNN++ on $8 \times 8$ down-sampled image. The second layer is a conditional PixelCNN++ that generates $32 \times 32$ images conditioned on $8 \times 8$ images. The final layer generates $128 \times 128$ images conditioned on $32 \times 32$ images.

## 7.2 DECODER ARCHITECTURES

### 7.2.1 MNIST

**ResBlock Architecture**

| Conv2d (channels, $k = 1$, no bias) |
| :---: |
| BatchNorm, ReLU |
| Conv2d (channels, $k = 3$, padding 1, no bias) |
| BatchNorm, ReLU |
| Conv2d (channels, $k = 1$, no bias) |
| BatchNorm, ReLU |
| Shortcut Connection, ReLU |

**Decoder Architecture**

| $x \in \mathbb{R}^n$ |
| :---: |
| Linear (1024), ReLU |
| ConvTranspose (128 filters, $k = 5$, stride 2), ReLU |
| ResBlock(128) |
| ConvTranspose (32 filters, $k = 5$, stride 2) |
| BatchNorm, ReLU |
| ConvTranspose (2 filter, $k = 5$, stride 2) |

### 7.2.2 CELEBA

**Convolutional Model**

| $x \in \mathbb{R}^{1 \times 1 \times 4096}$ |
| :---: |
| ConvTranspose (256 filters, $k = 4$) |
| 4x ResBlock, Upsample 2x |
| BatchNorm, ReLU |
| Conv2d (32 filters, $k = 1$) |

**WGAN**

Follows the same architecture described in https://github.com/LynnHo/DCGAN-LSGAN-WGAN-GP-DRAGAN-Pytorch. To make it conditional, we concatenate the conditioning vector with the noise when generating. For the discriminator, we project the conditioning vector to the channel dimension of each ResBlock input, and add it broadcasted across the image as a bias.

**RealNVP**

We use the Multiscale RealNVP architecture described in Dinh et al. (2016). We apply conditioning as follows: for each ResNet in the affine coupling layers, we project the conditioning vector to the channel dimension size, and add it as a bias to the ResNet input.

### 7.3 RECONSTRUCTIONS FOR DIFFERENT DECODERS

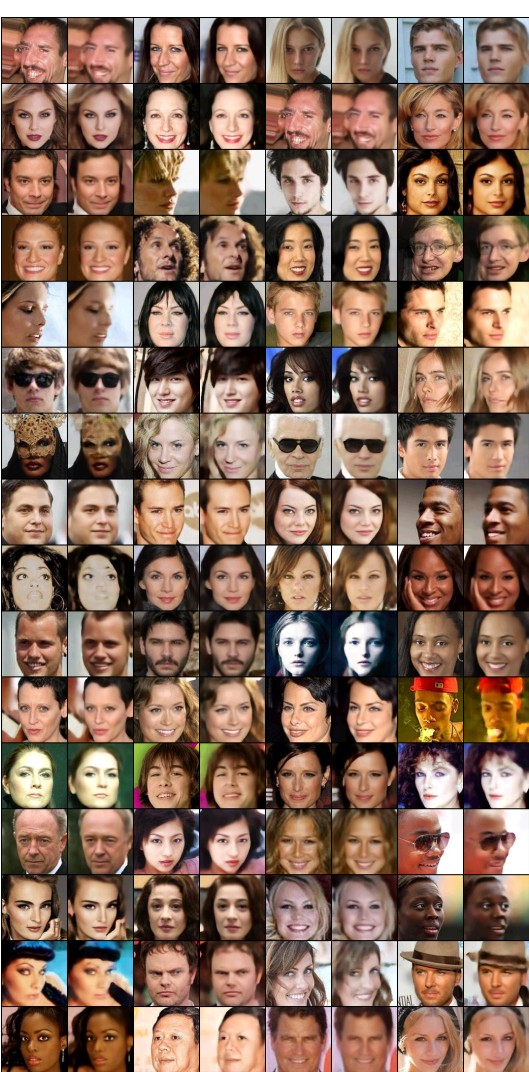

Figure 8: **Autoregressive model**: Gated PixelCNN, **Embedding Space**: Activations, **Decoder**: Convolutional Network

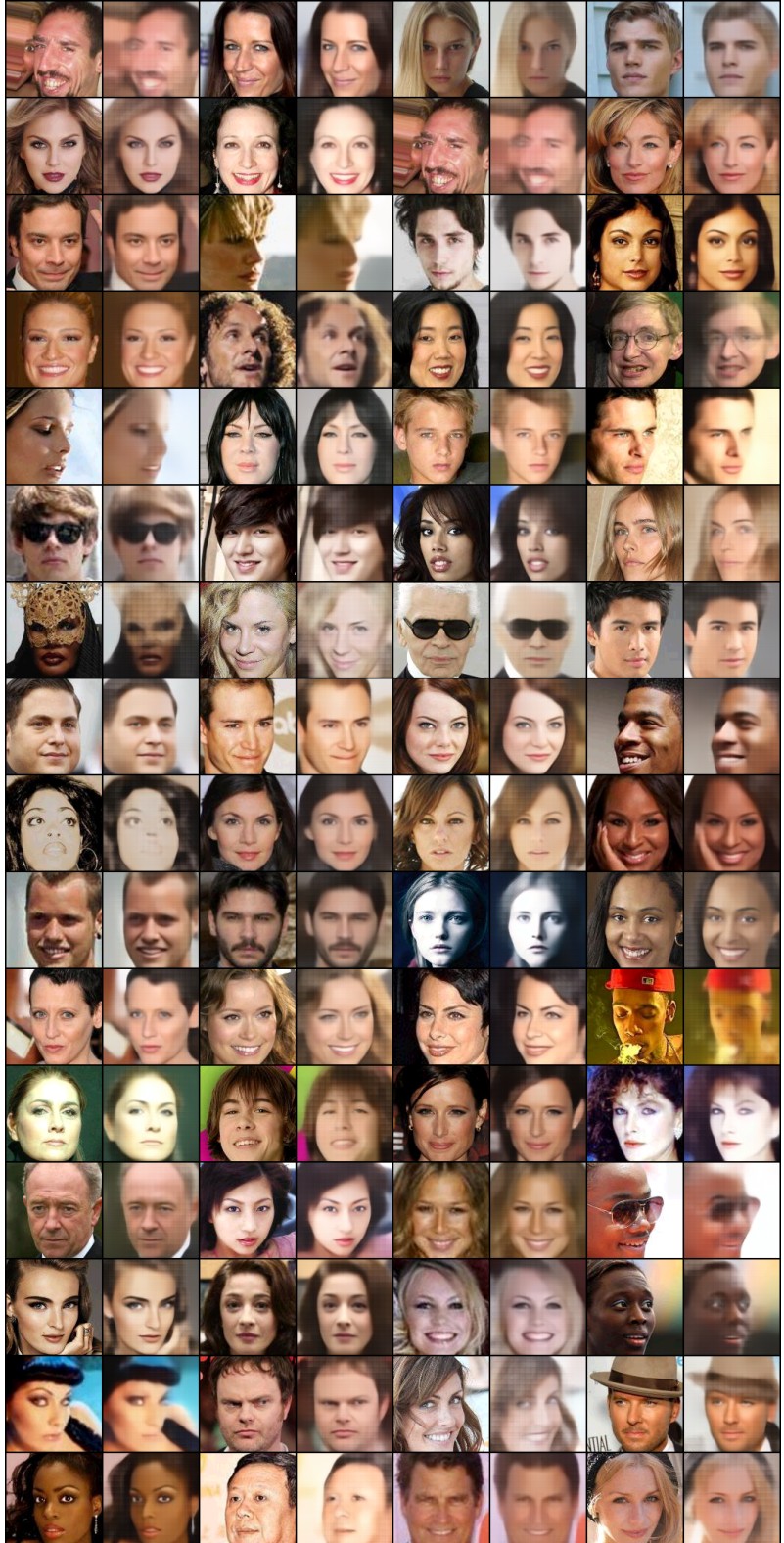

Figure 9: **Autoregressive model**: Gated PixelCNN, **Embedding Space**: Activations, **Decoder**: RealNVP

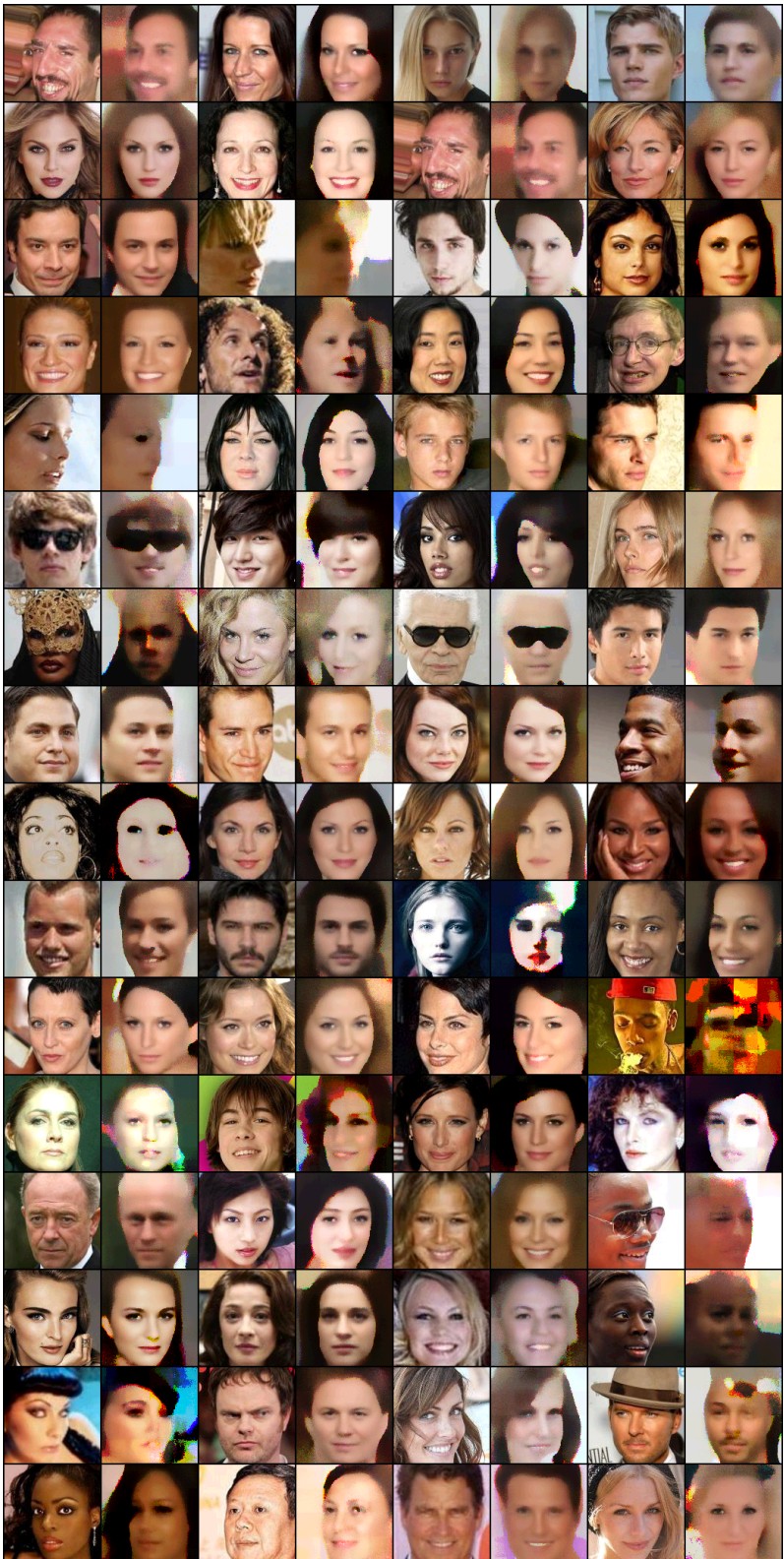

Figure 10: **Autoregressive model**: Gated PixelCNN, **Embedding Space**: Fisher Scores, **Decoder**: Convolutional Network

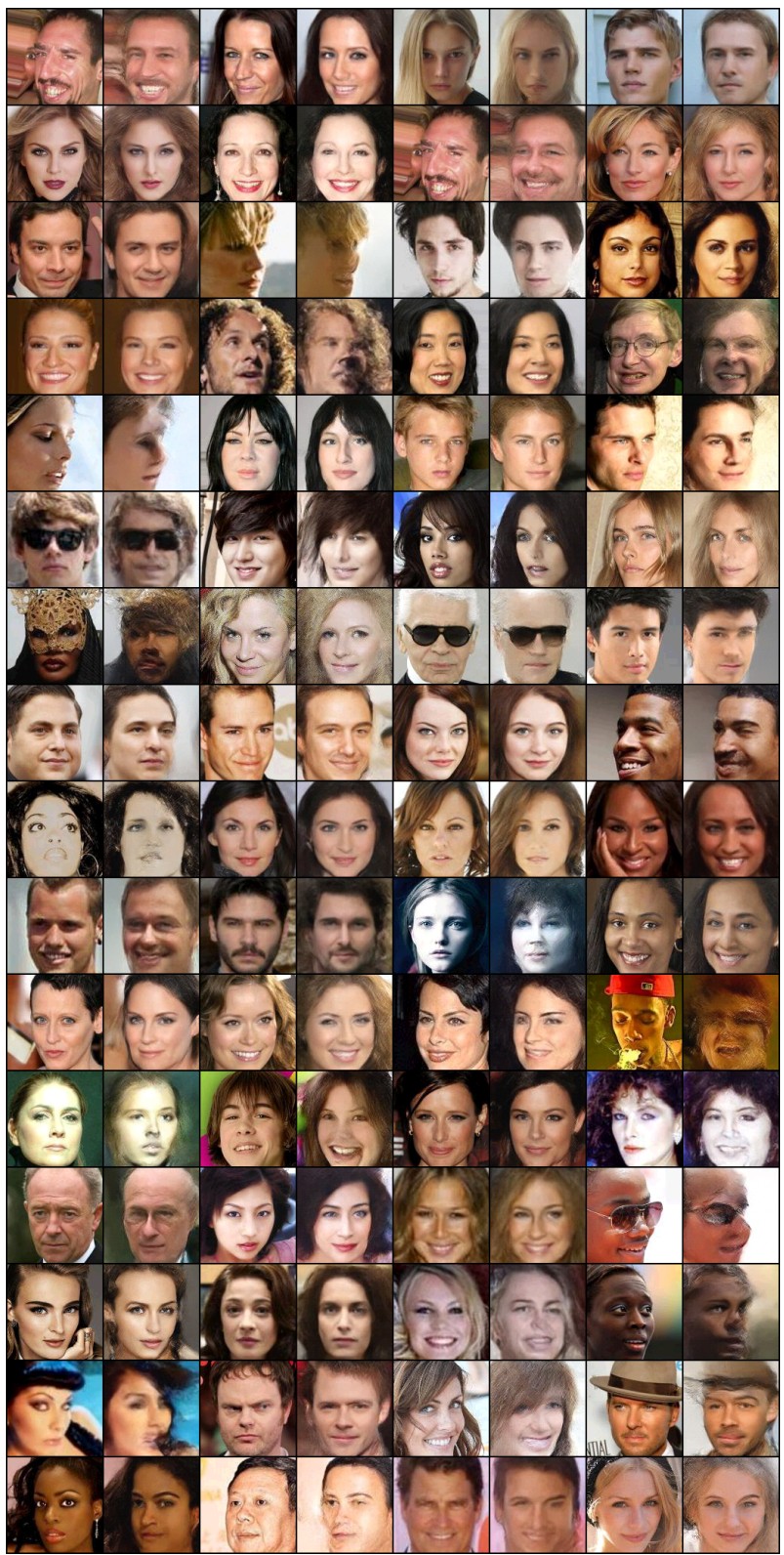

Figure 11: **Autoregressive model**: Gated PixelCNN, **Embedding Space**: Fisher Scores, **Decoder**: Conditional RealNVP

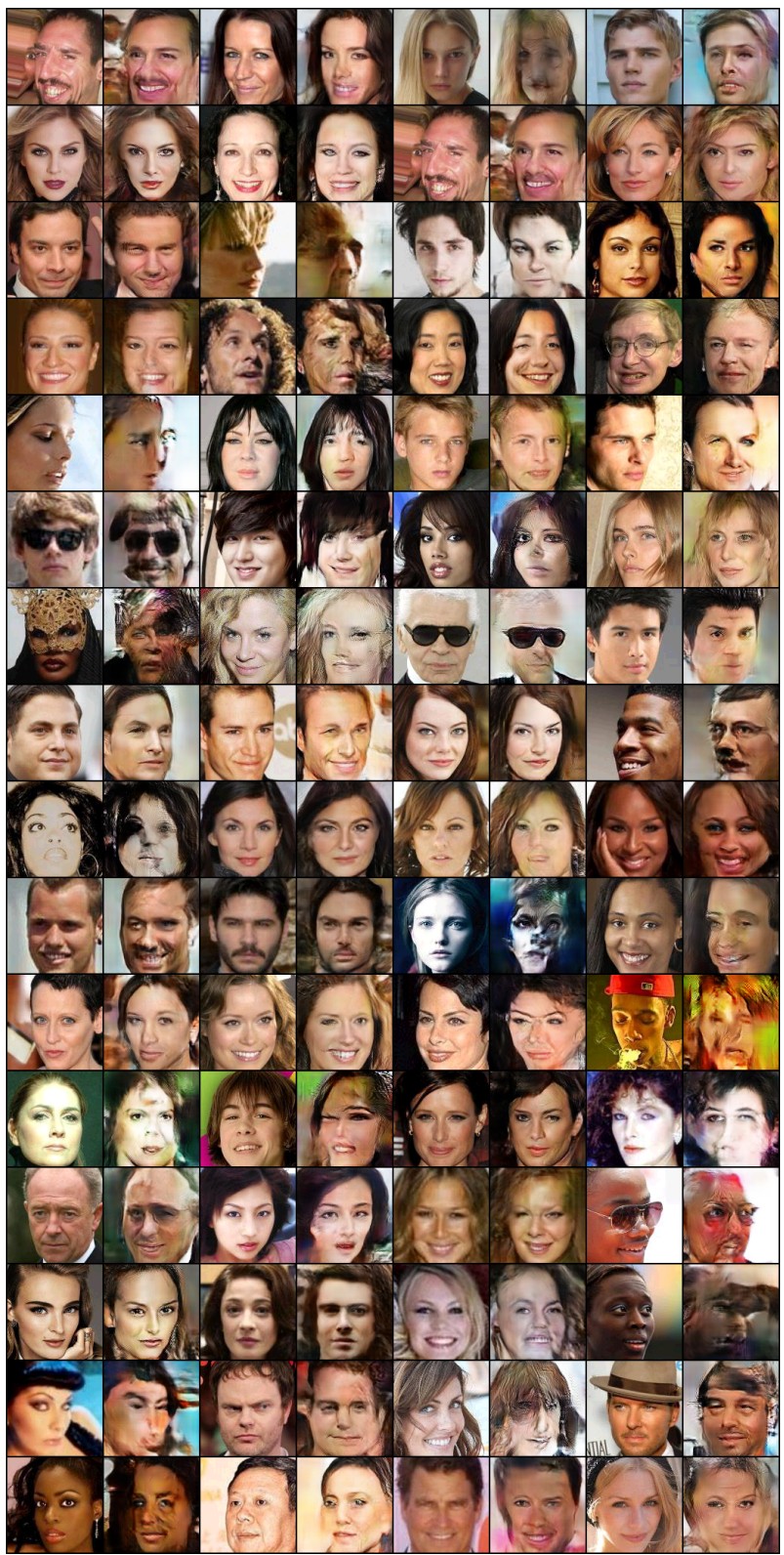

Figure 12: **Autoregressive model**: Gated PixelCNN, **Embedding Space**: Fisher Scores, **Decoder**: Conditional WGAN

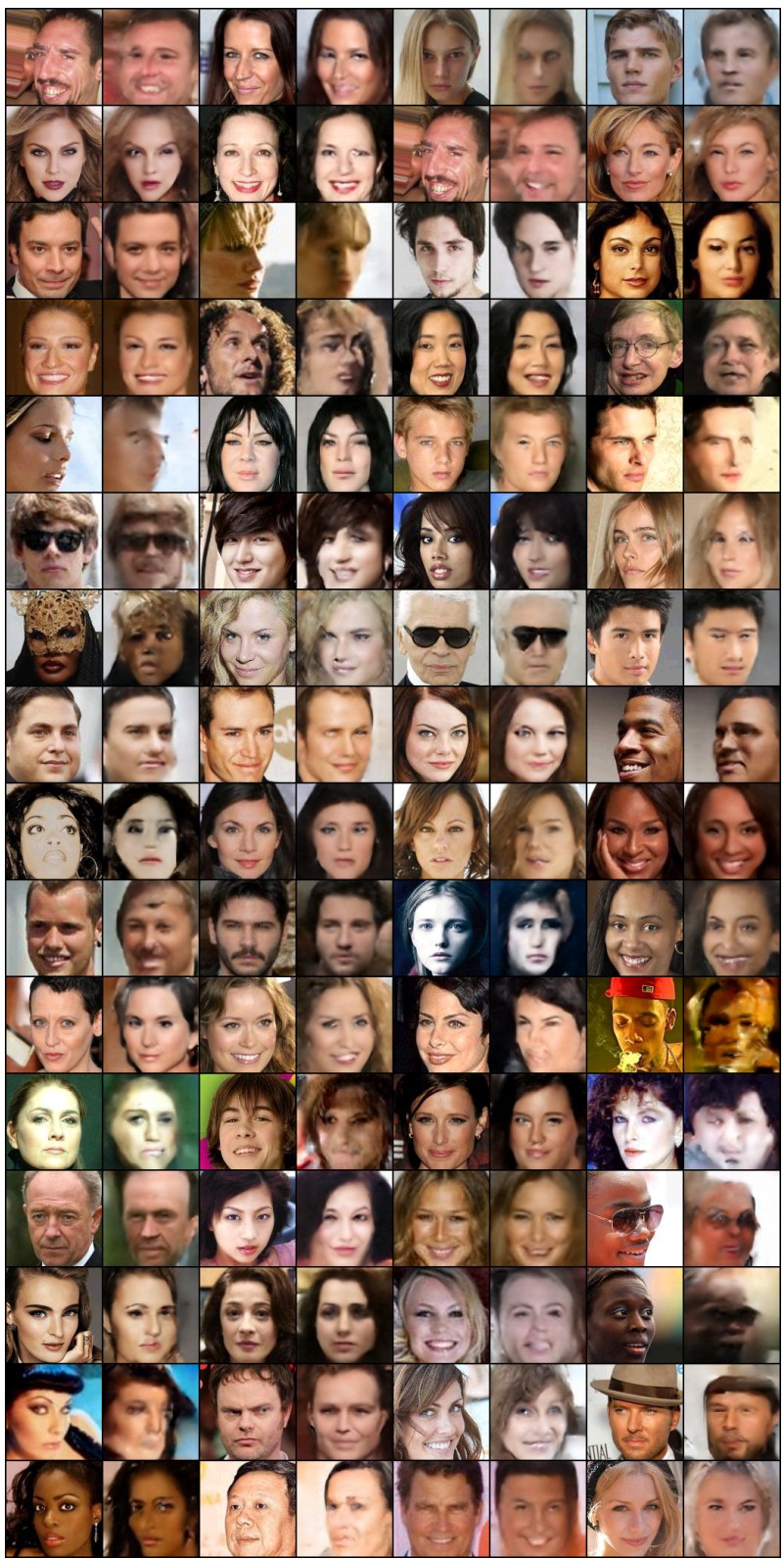

Figure 13: **Autoregressive model**: Gated PixelCNN, **Embedding Space**: Fisher Scores, **Decoder**: Conditional Pyramid PixelCNN

## 7.4 INTERPOLATIONS FOR DIFFERENT DECODERS

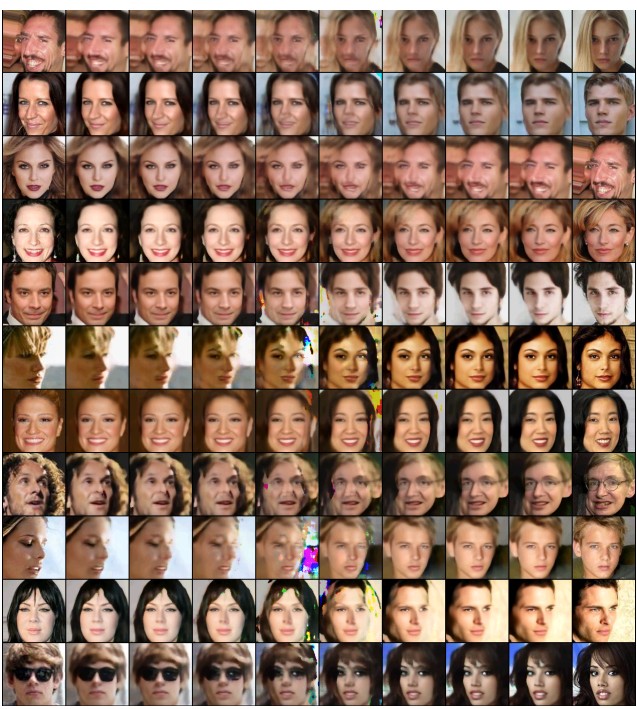

(a) **Autoregressive model**: Gated PixelCNN, **Embedding Space**: Activations, **Decoder**: Convolutional Network

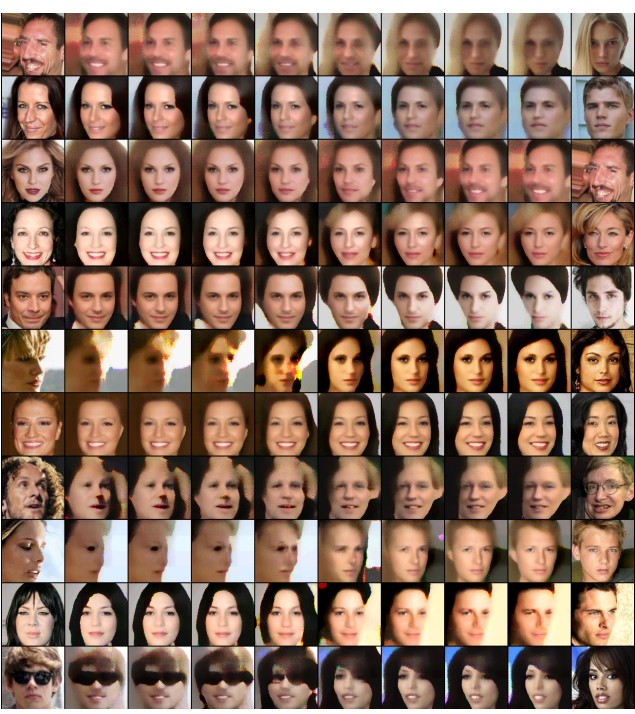

(b) **Autoregressive model**: Gated PixelCNN, **Embedding Space**: Fisher Scores, **Decoder**: Convolutional Network

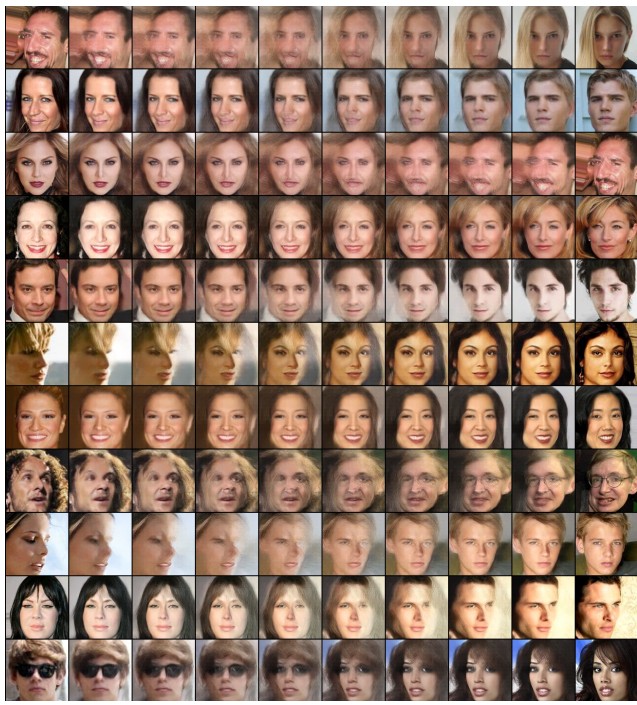

(a) **Autoregressive model**: Gated PixelCNN, **Embedding Space**: Activations, **Decoder**: RealNVP

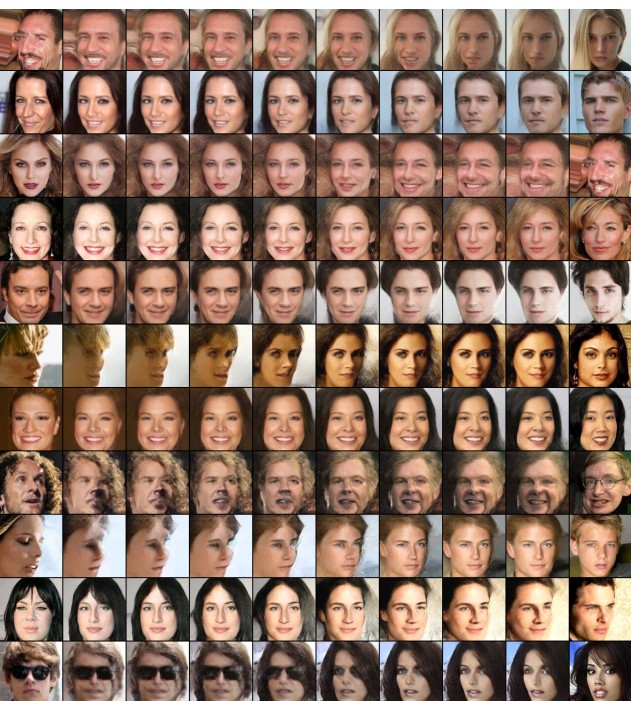

(b) **Autoregressive model**: Gated PixelCNN, **Embedding Space**: Fisher Scores, **Decoder**: Conditional Real-NVP

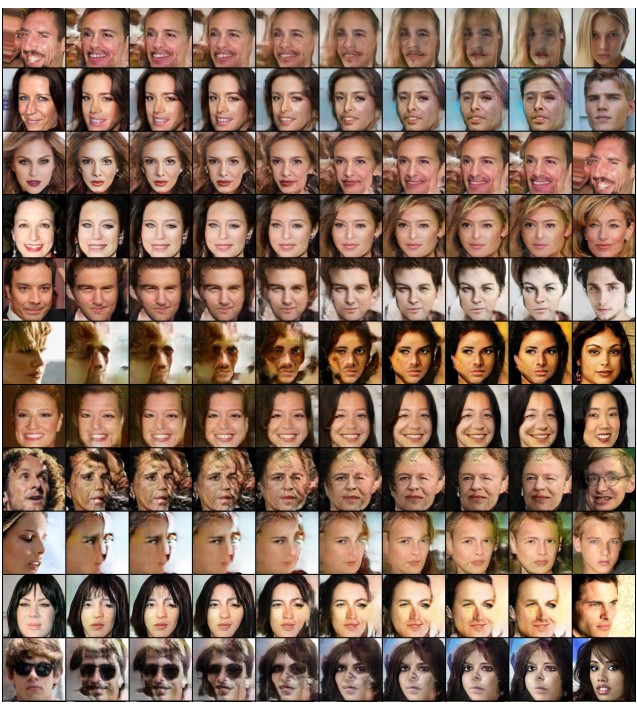

(a) **Autoregressive model**: Gated PixelCNN, **Embedding Space**: Fisher Scores, **Decoder**: Conditional WGAN

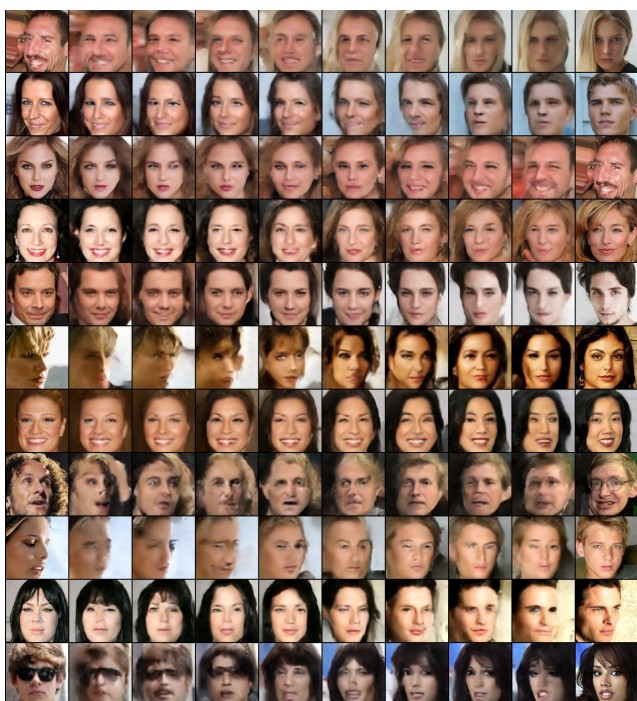

(b) **Autoregressive model**: Gated PixelCNN, **Embedding Space**: Fisher Scores, **Decoder**: Conditional Pyramid PixelCNN

7.5    FACIAL ATTRIBUTE MANIPULATION FOR DIFFERENT DECODERS



(a) **Autoregressive model**: Gated Pixel-CNN, **Embedding Space**: Activations, **Decoder**: Conditional RealNVP

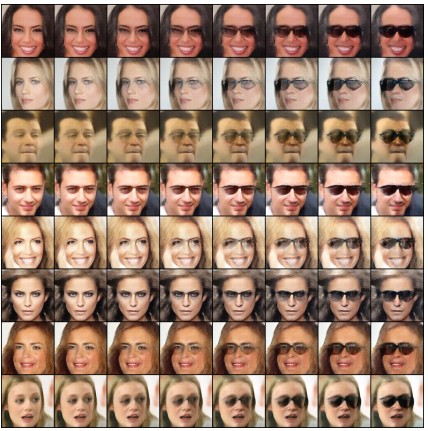

(b) **Autoregressive model**: Gated Pixel-CNN, **Embedding Space**: Activations, **Decoder**: Conditional RealNVP

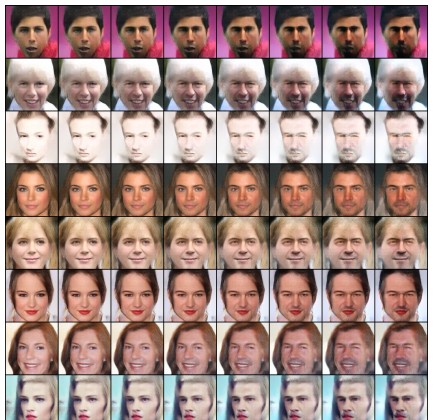

(c) **Autoregressive model**: Gated Pixel-CNN, **Embedding Space**: Activations, **Decoder**: Conditional RealNVP

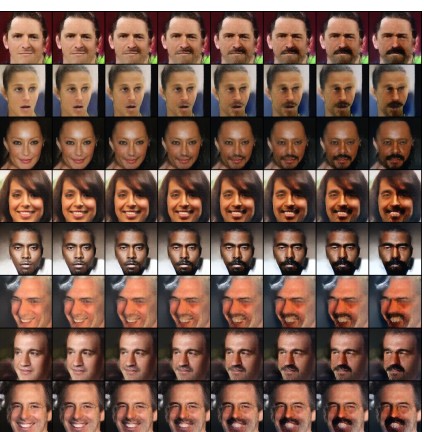

(d) **Autoregressive model**: Gated Pixel-CNN, **Embedding Space**: Activations, **Decoder**: Conditional RealNVP

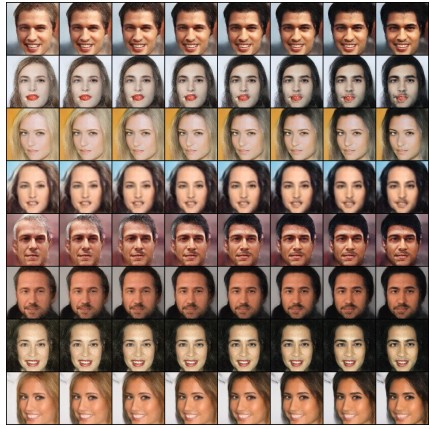

(a) **Autoregressive model**: Gated Pixel-CNN, **Embedding Space**: Fisher Score, **Decoder**: Conditional RealNVP

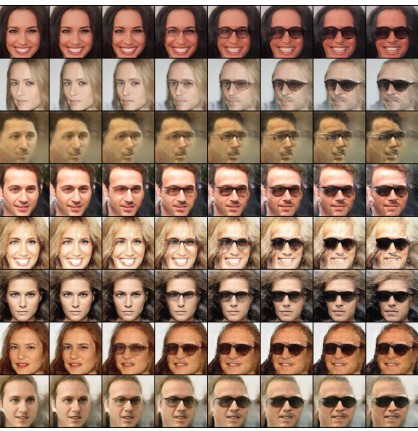

(b) **Autoregressive model**: Gated Pixel-CNN, **Embedding Space**: Fisher Score, **Decoder**: Conditional RealNVP

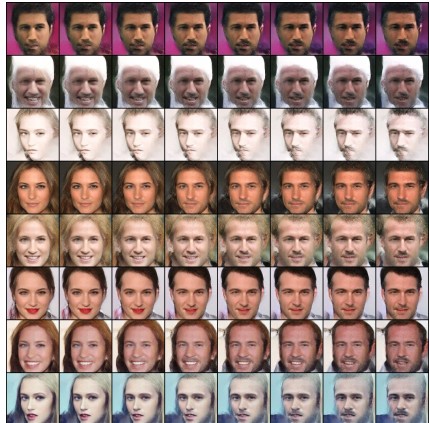

(c) **Autoregressive model**: Gated Pixel-CNN, **Embedding Space**: Fisher Score, **Decoder**: Conditional RealNVP

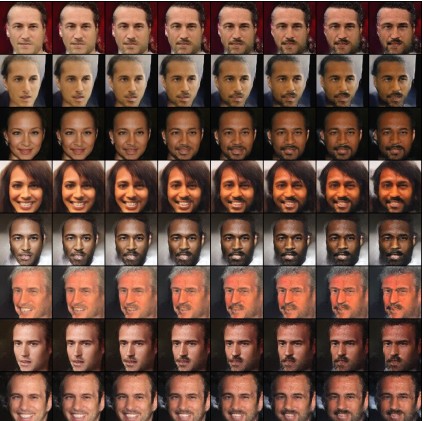

(d) **Autoregressive model**: Gated Pixel-CNN, **Embedding Space**: Fisher Score, **Decoder**: Conditional RealNVP

