# OpenReview forum: "Natural Image Manipulation for Autoregressive Models Using Fisher Scores"
_ICLR.cc/2020/Conference — Reject_

### Official Review · AnonReviewer3 · 2019-10-25
**Official Blind Review #3**

**Rating:** 8

**Review:**

This paper focuses on the problem of interpolating between data points using neural autoregressive models. The core idea is that it is possible to use (a smaller-dimensional projection of) the Fisher score of the density function defined by the autoregressive model to represent data points in embedding space, and a neural decoder for mapping them back to input space. Experiments on both MNIST and Celeb suggest that this is a sensible method, and leads to smoother interpolations rather than just relying on the embeddings resulting from the network activations.

Minor: the FID acronym on pg. 2 was not introduced beforehand.


**Experience Assessment:**

I do not know much about this area.

**Review Assessment: Checking Correctness Of Derivations And Theory:**

I carefully checked the derivations and theory.

**Review Assessment: Checking Correctness Of Experiments:**

I assessed the sensibility of the experiments.

**Review Assessment: Thoroughness In Paper Reading:**

I read the paper at least twice and used my best judgement in assessing the paper.

---

### Official Review · AnonReviewer1 · 2019-10-26
**Official Blind Review #1**

**Rating:** 3

**Review:**

Motivated by the observation that powerful deep autoregressive models such as PixelCNNs lack the ability to produce semantically meaningful latent embeddings and generate visually appealing interpolated images by latent representation manipulations, this paper proposes using Fisher scores projected to a reasonably low-dimensional space as latent embeddings for image manipulations. A decoder based on a CNN, a Conditional RealNVP, or a Conditional Pyramid PixelCNN is used to decode high-dimensional images from these projected Fisher score.  Experiments with different autoregressive and decoder architectures are conducted on MNIST and CelebA datasets are conducted.

Pros:

This paper is well-written overall and the method is clearly presented.


Cons:

1) It is well-known that the latent activations of deep autoregressive models don’t contain much semantically meaningful information. It is very obvious that either a CNN decoder, a conditional RealNVP decoder, or a conditional Pyramid PixelCNN decoder conditioned on projected Fisher scores will produce better images because the Fisher scores simply contain much more information about the images than the latent activations. When the $\alpha$ is small, the learned decoder will function similarly to the original pixelCNN, therefore, latent activations produce smaller FID scores than projected Fisher scores for small $\alpha$’s. These results are not surprising. Detailed explanations should be added here.

2) The comparisons to baselines are unfair. As mentioned in 1), it’s obvious that Fisher scores contain more information than latent activations for deep autoregressive models and are better suited for manipulations. Fair comparisons should be performed against other latent variable models such as flow models and VAEs with more interesting tasks, which will make the paper much stronger.

3) In Figure 3, how is the reconstruction error calculated? It’s squared error per pixel per image?

4) On pp. 8, for semantic manipulations, some quantitative evaluations will strengthen this part.

In summary, this paper proposes a novel method based on projected Fisher scores for performing semantically meaningful image manipulations under the framework of deep autoregressive models. However, the experiments are not well-designed and the results are unconvincing. I like the idea proposed in the paper and strongly encourage the authors to seriously address the raised questions regarding experiments and comparisons.

------------------
After Rebuttal:

I took back what I said. It's not that obvious that the "latent activations of deep autoregressive models don’t contain much semantically meaningful information". But the latent activations are indeed a weak baseline considering that PixelCNN is so powerful a generator. If the autoregressive generator is powerful enough, the latent activations can theoretically encode nothing.  I have spent a lot of time reviewing this paper and related papers, the technical explanation about the hidden activation calculation of PixelCNN  used in this paper is unclear and lacking (please use equations not just words).

Related paper:  The PixelVAE paper ( https://openreview.net/pdf?id=BJKYvt5lg ) explains that PixelCNN doesn't learn a good hidden representation for downstream tasks

Another paper combining VAE and PixelCNN also mentions this point:

ECML 2018: http://www.ecmlpkdd2018.org/wp-content/uploads/2018/09/455.pdf

Please also check the related arguments about PixcelCNN (and the "Unconditional Decoder" results) in Variational Lossy Autoencoder (https://arxiv.org/pdf/1611.02731.pdf )

As I mentioned in the response to the authors' rebuttal, training a separate powerful conditional generative model from some useful condition information (Fisher scores) is feasible to capture the global information in the condition, which is obvious to me. This separate powerful decoder has nothing to do with PixelCNN, which is the major reason that I vote reject.


**Experience Assessment:**

I have published one or two papers in this area.

**Review Assessment: Checking Correctness Of Derivations And Theory:**

I carefully checked the derivations and theory.

**Review Assessment: Checking Correctness Of Experiments:**

I carefully checked the experiments.

**Review Assessment: Thoroughness In Paper Reading:**

I read the paper thoroughly.

---

> ### Author Response · Authors · 2019-11-09
> **Response to Reviewer 1**
>
> Thank you for your review.
>
> > 1) It is well-known that the latent activations of deep autoregressive models don’t contain much semantically meaningful information.
>
> To the best of our knowledge, we do not know of any existing work that shows this property in deep autoregressive models. We would appreciate if you could provide references to such existing work.
>
> We would like to emphasize that our paper targets the problem of image manipulation with autoregressive models, and is not aiming to solve a representation learning problem. Therefore, we designed our experiments to show that Fisher scores are a better latent space than alternative latents in autoregressive models. Among possible latent spaces that could be extracted from autoregressive models, we found layer activations to be the strongest candidate. We believe such a choice is justified, as layer activations are also commonly used in many prior self-supervised learning methods [3] [4].
>
> > 2) It’s obvious that Fisher scores contain more information than latent activations for deep autoregressive models and are better suited for manipulations
>
> We agree that Fisher scores do contain more information than latent activations. However, we believe that it is not obvious that projected Fisher scores provides a latent space which entails more meaningful semantic manipulations for global attributes such as pose, hair color, and lighting. Experiments in [2] showed that smaller receptive fields attain competitive log-likelihood scores, suggesting that PixelCNNs model low-level statistics about the data. However, our experiments with using Fisher scores as an embedding space to interpolate show otherwise to common knowledge.
>
> > comparisons should be performed against other latent variable models
>
> Our paper aims to show that image manipulation is possible using autoregressive models, despite the lack of easily accessible latents such as those in VAEs or flow models. We believe that it is more meaningful to compare our method against existing and alternative methods that perform image manipulation using an autoregressive model, which motivates our decision as using layer activations for our baseline method.
>
> > 3) how is reconstruction error calculated
>
> The decoder was trained to model discrete pixels values, so the reconstruction error is negative log-likelihood (nats per dim).
>
> > 4) for semantic manipulations, some quantitative evaluations will strengthen this part
>
> To the best of our knowledge, we do not know of a method to effectively quantitatively evaluate the quality of semantic manipulations. We experimented with a few different evaluation metrics, such as using binary classifiers in a method similar to [1], however, none showed meaningful results as an evaluation metric. Training binary classifiers on CelebA attributes proved too easy, and provided no discernable difference between semantic manipulations using different embedding spaces even when there was a clear visual difference. We were also unable to use the same FID evaluating metric as interpolations, since semantic manipulations by design conditionally generate images out of distribution, whereas FID compares datasets of the same distribution.
>
> [1] Ravuri, Suman, and Oriol Vinyals. "Classification Accuracy Score for Conditional Generative Models." arXiv preprint arXiv:1905.10887 (2019).
> [2] Salimans, Tim, et al. "Pixelcnn++: Improving the pixelcnn with discretized logistic mixture likelihood and other modifications." arXiv preprint arXiv:1701.05517 (2017).
> [3] Noroozi, Mehdi, and Paolo Favaro. "Unsupervised learning of visual representations by solving jigsaw puzzles." European Conference on Computer Vision. Springer, Cham, 2016.
> [4] Gidaris, Spyros, Praveer Singh, and Nikos Komodakis. "Unsupervised representation learning by predicting image rotations." arXiv preprint arXiv:1803.07728 (2018).

---

> > ### Comment · AnonReviewer1 · 2019-11-13
> > **Thanks for the response**
> >
> > "> The paper uses the activation of the last layers of the PixelCNN as a baseline, which I consider to be a very weak baseline:   Our paper focuses on the question: is it possible to perform image interpolation using autoregressive models? Our experiments were designed to show that project Fisher scores provide better and more meaningful image manipulations than any other alternative latent spaces inherent to an autoregressive model, such as layer activations and noise space. In particular, the goal of our work is not to develop a new representation learning method; it is to develop an image interpolation method using autoregressive models. We believe that this is an interesting avenue of research because autoregressive models are not clearly suited for image interpolation, unlike VAEs and flows which have a clearly usable latent space."
> >
> > The response from the authors to Review #4 is also highly related to my question. My major concern still remains. Since PixelCNN can have impressive performance as a generative model, it is really unsurprising that it can capture global semantic information considering its parameters as a whole and the associated Fisher scores. This paper just wants to prove this unsurprising point using a separate powerful decoder.
> >
> > The authors also argue that "Our paper does not focus on solving unsupervised learning using Fisher scores. Rather, it focuses on a novel method using Fisher scores to perform image manipulation using autoregressive models. We believe that a metric designed to evaluate interpolations is a more direct quantitative measurement of image manipulation quality, compared to standard representation learning evaluation methods."
> >
> > I still feel that this argument is not enough to convince me that showing the parameters of PixelCNN as a whole can capture global semantic information is interesting.
> >
> > To summarize, from a reviewer's perspective, showing A can do B is not interesting (it's unsurprising); many other C, D, E, F, G can do B too; if you want to convince me, it's better to show that A can do B much better than C, D, E, F, G, which might seem to be controversial. But indeed, training a powerful conditional generative model from some useful condition information is feasible to capture the global information in the condition, which is obvious to me.

---

### Official Review · AnonReviewer4 · 2019-11-02
**Official Blind Review #4**

**Rating:** 1

**Review:**

This work proposes to learn a latent space for the PixelCNN by first computing the Fisher score of the PixelCNN model and then projecting it onto a lower-dimensional space using a sparse random matrix.

My first concern about this work is its novelty. Defining a feature space using the Fisher kernel of a generative model is a very well-known idea, and there is a large body of work around that. As the paper points out, the problem with the Fisher score for the recent deep generative model architectures is that Fisher score operates in the parameter space and the deep models have a very large number of parameters. The paper proposes to get around this problem by projecting the Fisher score onto a lower-dimensional space using random matrices. But I am not convinced that this random projection can learn useful representations, which brings me to my second concern about the evaluation metric. The paper uses the activation of the last layers of the PixelCNN as a baseline, which I consider to be a very weak baseline. Each activation at spatial position (i,j) only depends on the previous pixels and I believe they are not in general good high-level representations. For the evaluation, the paper only considers the FID score on the interpolated images and reconstructions. There are much better ways to compare the quality of unsupervised representations such as their performance on classifying images with a linear classifier as done in [1]. The paper would improve by comparing the quality of its latent representations with the recent unsupervised/self-supervised learning methods such as [1,2].

[1] Data-Efficient Image Recognition with Contrastive Predictive Coding
[2] Learning deep representations by mutual information estimation and maximization

**Experience Assessment:**

I have published in this field for several years.

**Review Assessment: Checking Correctness Of Derivations And Theory:**

N/A

**Review Assessment: Checking Correctness Of Experiments:**

I assessed the sensibility of the experiments.

**Review Assessment: Thoroughness In Paper Reading:**

I read the paper at least twice and used my best judgement in assessing the paper.

---

> ### Author Response · Authors · 2019-11-09
> **Response to Reviewer 4**
>
> Thank you for your review. We would like to emphasize that the novelty of our work is showing the Fisher scores of PixelCNNs do indeed contain high-level semantic information about the data, which we believe is interesting given the common knowledge that PixelCNNs tend to model only low-level statistics of the data. Prior experiments such as in [2] showed that PixelCNNs with small receptive fields maintain log-likelihood performance close to PixelCNNs with large receptive fields, suggesting that PixelCNNs focus on modeling low level statistics about the data, rather than global semantically meaningful information.
>
> We show that contrary to this common knowledge, PixelCNNs actually do contain global semantically meaningful information — although to access this information one must be careful to extract it in the correct way.  Our experiments show that naively interpolating using PixelCNN layer activations is not the correct way, as it results in non-meaningful interpolations. Surprisingly, however, we show that interpolating in Fisher score space does result in semantically meaningful interpolations. We believe that this demonstrates that PixelCNNs do capture high-level information about the data, contrary to the common knowledge that they only capture low-level statistics.
>
> > The paper uses the activation of the last layers of the PixelCNN as a baseline, which I consider to be a very weak baseline
>
> Our paper focuses on the question: is it possible to perform image interpolation using autoregressive models? Our experiments were designed to show that project Fisher scores provide better and more meaningful image manipulations than any other alternative latent spaces inherent to an autoregressive model, such as layer activations and noise space. In particular, the goal of our work is not to develop a new representation learning method; it is to develop an image interpolation method using autoregressive models. We believe that this is an interesting avenue of research because autoregressive models are not clearly suited for image interpolation, unlike VAEs and flows which have a clearly usable latent space.
>
> > For the evaluation, the paper only considers the FID score on the interpolated images and reconstructions. There are much better ways to compare the quality of unsupervised representations …
>
> We designed our evaluation metric with the intention to quantitatively show that interpolations using Fisher scores are better than those of using alternative embedding spaces of an autoregressive model. Empirically, using FID on interpolated datasets showed a strong correlation between FID scores and interpolation quality.
>
> Our paper does not focus on solving unsupervised learning using Fisher scores. Rather, it focuses on a novel method using Fisher scores to perform image manipulation using autoregressive models. We believe that a metric designed to evaluate interpolations is a more direct quantitative measurement of image manipulation quality, compared to standard representation learning evaluation methods.

---

### Comment · Area_Chair1 · 2019-11-13
**Thanks for your reviews. Please take a look at the rebuttal.**

Dear reviewers,

Thank you very much for your efforts in reviewing this paper.

The authors have provided their rebuttal. It would be great if you take a look at them, and see whether it changes your opinion in anyway. If there is still any unclear point or a serious disagreement, please bring it up. Also if you are hoping to see a specific change or clarification in the paper before you update your score, please mention it.

The authors have only until November 15th to reply back.

I also encourage you to take a look at each others’ reviews. There might be a remark in other reviews that changes your opinion.

Thank you,
Area Chair

---

### Decision · Program_Chairs · 2019-12-19

**Decision:**

Reject

**Comment:**

The paper proposes learning a latent embedding for image manipulation for PixelCNN by using Fisher scores projected to a low-dimensional space.
The reviewers have several concerns about this paper:
* Novelty
* Random projection doesn’t learn useful representation
* Weak evaluations
Since two expert reviewers are negative about this paper, I cannot recommend acceptance at this stage.